# Salivary and Serum Interferon-Gamma/Interleukin-4 Ratio in Oral Lichen Planus Patients: A Systematic Review and Meta-Analysis

**DOI:** 10.3390/medicina55060257

**Published:** 2019-06-08

**Authors:** Hamid Reza Mozaffari, Maryam Molavi, Pia Lopez-Jornet, Masoud Sadeghi, Mohsen Safaei, Mohammad Moslem Imani, Roohollah Sharifi, Hedaiat Moradpoor, Amin Golshah, Ladan Jamshidy

**Affiliations:** 1Department of Oral and Maxillofacial Medicine, School of Dentistry, Kermanshah University of Medical Sciences, Kermanshah 6713954658, Iran; mozaffari20@yahoo.com; 2Medical Biology Research Center, Kermanshah University of Medical Sciences, Kermanshah 6714415185, Iran; 3Students Research Committee, Kermanshah University of Medical Sciences, Kermanshah 6715847141, Iran; maryam.molavi72@gmail.com; 4Facultad de Medicina y Odontologia Universidad de Murcia, Hospital Morales Meseguer, Clinica Odontologic Adv Marques Velez s/n, 30008 Murcia, Spain; majornet@um.es; 5Oral and Dental Sciences Research Laboratory, School of Dentistry, Kermanshah University of Medical Sciences, Kermanshah 6713954658, Iran; mohsen_safaei@yahoo.com; 6Department of Orthodontics, School of Dentistry, Kermanshah University of Medical Sciences, Kermanshah 6713954658, Iran; mmoslem.imani@yahoo.com (M.M.I.); amin.golshah@gmail.com (A.G.); 7Department of Endodontics, School of Dentistry, Kermanshah University of Medical Sciences, Kermanshah 6713954658, Iran; roholahsharifi@gmail.com; 8Department of Prosthodontics, School of Dentistry, Kermanshah University of Medical Sciences, Kermanshah 6713954658, Iran; hedaiat.moradpoor@gmail.com (H.M.); ladanjamshidy@yahoo.com (L.J.)

**Keywords:** oral lichen planus, ratio, interferon-gamma, interleukin-4, meta-analysis

## Abstract

*Background and Objectives:* Interferon-gamma (IFN-γ)/interleukin-4 (IL-4) ratio may indicate a change in the immune response with a potential pathological effect presented in oral lichen planus (OLP) patients. Herein, this meta-analysis evaluated the role of serum and salivary interferon-gamma/interleukin-4 ratio in the severity and development of OLP. *Materials and Methods:* The Scopus, Cochrane Library, PubMed, and Web of Science databases were systematically searched to retrieve the relevant studies published up from the database inception to March 2019. The crude mean difference (MD) and 95% confidence interval (CI) were calculated by RevMan 5.3 software using a random-effects model. A sensitivity analysis was performed on the results using the CMA 2.0 software. A total of 98 studies were retrieved from the databases, of which at last seven studies were included in this meta-analysis. *Results:* The findings showed that the pooled MDs of serum and salivary IFN-γ/IL-4 ratio were −0.22 (95% CI: −1.16, 0.72; *p* = 0.64) and 0.17 (95% CI: −1.50, 1.84; *p* = 0.84) in OLP patients compared to controls, respectively. In addition, the pooled MDs of serum and salivary IFN-γ/IL-4 ratio were −0.15 (95% CI: −0.53, 0.23; *p* = 0.43) and −0.39 (95% CI: −0.63, −0.15; *p* = 0.001) in patients with erythematous/ulcerative subtype compared to patients with reticular subtype, respectively. *Conclusions:* In conclusion, the results of meta-analysis demonstrated that serum and salivary IFN-γ/IL-4 ratio cannot play a major role in OLP development and severity.

## 1. Introduction

Oral lichen planus (OLP) is a chronic inflammatory disorder of oral mucosa that affects 0.5% to 4% of the general population [1], two to three times more frequently in females than in males [2], especially in females over the age of 40 years [3]. The OLP etiology is still unclear [3,4], but additional evidence suggests an immune response to T cells against epithelial cells [4]. OLP is associated with various other systemic diseases and conditions [5]. This disease has several clinical manifestations including reticular, plaque-like, papular, bullous, erythematous, and erosive/ ulcerative forms [6]. The reticular is the most common type of OLP and yet it is characterized by low/moderate immune response compared with other forms, whereas the erosive and erythematous are less common and cause more serious symptoms [7]. T helper (Th) cells are classified into two subtypes (Th1 and Th2) according to cytokine production [8]. Interferon-gamma (IFN-γ) is a soluble dimer cytokine, also called active macrophage factor [9], which causes an inflammatory response and apoptotic cell death [10]. This cytokine plays an important role in consistent and inherent immunity, particularly against tumor control, viral infection, and intracellular bacteria [11]. Interleukin-4 (IL-4) is a major immunomodulatory cytokine that is mainly included in adaptive immunity [12]. This interleukin is involved in the activation of B and T cells, humoral immune response, and reduction of pathological inflammation [13]. IFN-γ and IL-4 are assessed to be the characteristic cytokines created by Th1 and Th2 cells, respectively [14]. As IFN-γ prevents the expression of Th2 cytokines such as IL-4 and vice versa, the IFN-γ/IL-4 ratio is considered to be a straightforward indicator of Th1/Th2 balance [15]. The variation of IFN-γ, IL-4, or IFN-γ/IL-4 ratio may indicate a change in the immune response with a potential pathological effect presented in OLP patients [8,16]. The purpose of this meta-analysis was to evaluate the role of serum and salivary interferon-gamma/interleukin-4 ratio in the severity and development of OLP.

## 2. Materials and Methods

The Preferred Reporting Items for Systematic Reviews and Meta-Analyses (PRISMA) protocol was used to design the present meta-analysis [17]. 

### 2.1. Search Strategy

The Scopus, Cochrane Library, PubMed/Medline, and Web of Science databases were searched to retrieve the relevant studies published from the database inception to March 2019. The search terms were (“oral lichen planus” OR “OLP”) AND (“interleukin-4” OR “IL-4”) AND (“Interferon-gamma” OR “IFN-gamma” OR “IFN-γ” OR “Interferon-γ” OR “IFN” OR “Interferon”) without any restriction. In addition, we searched the references of the retrieved studies related to the topic to make sure no study was missed.

### 2.2. Study Selection

One author (M.S.) retrieved the articles in the databases. He assessed the titles and abstracts of the relevant articles and uploaded and screened the full-texts of the articles that met our eligibility criteria. After the full-text screening, the reason for exclusion was mentioned for any study excluded. Another author (H.R.M.) independently re-investigated the full-texts. The disagreements between the two authors were resolved by discussion.

### 2.3. Eligibility Criteria

The eligibility criteria were: (I) studies including both case (OLP patients) and healthy control groups and (II) studies reporting IFN-γ/IL-4 ratio in saliva and/or serum. Studies including just case group, trials, commentaries, letters to the editor, case reports, reviews, systematic reviews, and conference papers did not meet the eligibility criteria.

### 2.4. Data Extraction

The data was extracted from each study by one author (M.S.) and analyzed in the meta-analysis are presented in Table 1. Another author (M.M.) independently re-checked them and if there was an error, he corrected it.

### 2.5. Quality Assessment

The quality assessment of each study was performed by the Newcastle–Ottawa scale (NOS) [18]. A study with a score ≥7 had high quality. The quality evaluation was independently carried out by two authors (M.S. and M.M.I.) and the results of both authors were similar.

### 2.6. Statistical Analysis

The crude mean difference (MD) and 95% confidence interval (CI) were obtained for each study by Review Manager 5.3 (RevMan 5.3; The Cochrane Collaboration, Oxford, UK) to show the strength of the results. A Z test was applied to assess the significance of the pooled MD, with a *p*-value (2-tailed) of less than 0.05. Heterogeneity was estimated using the chi-squared (χ^2^, or Chi^2^) test, the Tau^2^ (the variance of the true effect sizes), and the I^2^ statistic such that *p* < 0.1 (I^2^ > 50%) showed a significant heterogeneity; therefore, we used the random-effects model. The funnel plot analysis was done using the Comprehensive Meta-Analysis version 2.0 (CMA 2.0; Biostat Inc, Englewood, NJ, USA) software using both Egger’s and Begg’s tests, *p* < 0.05 (2-tailed) being considered the significance degree of publication bias. We used the removal of one study, cumulative analysis, and omission of an outlier to evaluate the stability/consistency of the results.

## 3. Results

### 3.1. Study Selection

A total of 98 studies were retrieved from the databases (Figure 1). After excluding the duplicate studies, 42 studies were screened. Out of all screened studies, 30 non-relevant studies were excluded, among which the full-texts of 12 studies were assessed. Five studies were excluded with reasons during the full-text assessment (one study was without a control group, one study was merely an abstract, one study included IFN-γ mRNA/IL-4 mRNA ratio, one study reported the ratio in peripheral blood mononuclear cells, and one study reported the ratio in exosomes). At last, seven studies were analyzed in the present meta-analysis. In addition, we checked the references of original and review articles related to the subject in order to find possible missed studies.

### 3.2. Study Characteristics

We extracted the characteristics of seven studies involved in the meta-analysis (Table 1). The studies had been published from 2008–2018, from which six studies were from China [4,15,19,20,21,22] and one study from Iran [23]. Four studies [4,19,20,23] reported the ratio in saliva, two studies [15,21] in serum, and one study [22] in both saliva and serum. Out of all studies, the measurement method of IFN-γ and IL-4 for calculating the IFN-γ/IL-4 ratio was enzyme-linked immunosorbent assay (ELISA) in six studies [15,19,20,21,22,23] and cytometric bead array in one study [4].

### 3.3. Meta-Analysis Report

#### 3.3.1. OLP vs. Control (Serum)

Figure 2 shows the pooled MD of serum IFN-γ/IL-4 ratio in 131 patients with OLP compared to 94 controls, which was −0.22 (95% CI: −1.16, 0.72; *p* = 0.64; I^2^ = 95% (*p_h_* < 0.00001)). Therefore, the results did not show a significant difference between the two groups.

#### 3.3.2. OLP vs. Control (Saliva)

Figure 3 shows the pooled MD of salivary IFN-γ/IL-4 ratio in 262 patients with OLP compared to 165 controls, which was 0.17 (95% CI: −1.50, 1.84; *p* = 0.84; I^2^ = 99% (*p_h_* < 0.00001)). Therefore, the results showed no significant difference between the two groups.

Figure 4 shows the pooled MD of serum IFN-γ/IL-4 ratio in 60 patients with erythematous/ulcerative subtype compared to 36 patients with reticular subtype. The pooled MD was −0.15 (95% CI: −0.53, 0.23; *p* = 0.43; I^2^ = 85% (*p_h_* = 0.009)). Therefore, the results showed no significant difference between the two subtypes.

Figure 5 shows the pooled MD of salivary IFN-γ/IL-4 ratio in 141 patients with erythematous/ulcerative subtype compared to 80 patients with reticular subtype. The pooled MD was −0.39 (95% CI: −0.63, −0.15; *p* = 0.001; I^2^ = 75% (*p_h_* = 0.007)). Therefore, the results demonstrated a significant difference between the two subtypes, salivary IFN-γ/IL-4 ratio being higher in reticular subtype than erythematous/ulcerative subtype.

### 3.4. Sensitivity Analysis

Sensitivity analyses, including the removal of one study, and cumulative analysis were performed on the previous results, which did not change the results. Therefore, these sensitivity analyses showed the stability of the results. We could not do these sensitivity analyses on the serum IFN-γ/IL-4 ratio in the patients with erythematous/ulcerative subtype compared to patients with reticular subtype because there were less than three studies. In addition, we omitted one outlier of the salivary IFN-γ/IL-4 ratio in patients with OLP compared to controls and also salivary IFN-γ/IL-4 ratio in patients with erythematous/ulcerative subtype compared to patients with reticular subtype (Table 2). The results confirmed that salivary IFN-γ/IL-4 ratio was significantly higher in patients with reticular subtype than patients with erythematous/ulcerative subtype (*p* = 0.0009).

### 3.5. Quality Assessment

The quality assessment showed that all studies had high quality, with the mean quality of 7.7 (Table 3).

## 4. Discussion

Immunological mechanisms can play an important role in the pathogenesis of OLP [24]. Cytokine ratio is reviewed to be a straightforward indicator of Th1/Th2 balance [25]. The present study checked the serum and salivary ratio of IFN-γ/IL-4 in OLP patients compared to controls and also erythematous/ulcerative subtype compared to reticular subtype. The results showed no significant difference between groups except for the salivary ratio of IFN-γ/IL-4 in erythematous/ulcerative subtype compared to reticular subtype, the ratio being higher in reticular subtype than another subtype.

Out of three studies reporting the serum ratio of IFN-γ/IL-4 [15,21,22], two studies [21,22] showed a significantly decreased ratio and one study [15] a significantly elevated ratio in OLP patients compared to controls. Out of five studies reporting the salivary ratio of IFN-γ/IL-4 [4,19,20,22,23], two studies [20,22] showed a significantly decreased ratio and two studies [19,23] a significantly elevated ratio in OLP patients compared to controls. Comparing serum IFN-γ/IL-4 ratio between OLP subtypes in two studies [21,22], one study [22] showed a significantly decreased ratio in erythematous/ulcerative subtype compared to reticular subtype. Further, comparing salivary IFN-γ/IL-4 ratio between OLP subtypes in four studies [19,20,22,23], two studies [20,23] showed a significantly decreased ratio in erythematous/ulcerative subtype compared to reticular subtype. Therefore, some results indicated that Th1 cell is more dominant than the Th2 cell [23]. However, the meta-analysis showed no significant difference between groups except for the salivary ratio of IFN-γ/IL-4 in erythematous/ulcerative subtype compared to reticular subtype, the ratio being higher in reticular subtype than another. In addition, the analyses had a high heterogeneity and low studies included in each analysis ccould be one of the reasons for this heterogeneity. Therefore, the readers should pay attention to this point and further studies are needed to prove this difference.

Tao et al. [19] indicated the IFN-γ/IL-4 ratio could partly increase the IL-4 response, but the results did not support the hypothesis that Th1⁄Th2 imbalance is associated with the OLP development. However, the Th1 and Th2 responses coexist in OLP pathogenesis [26,27]. One study [21] identified that Golli-MBP (human Golli-myelin basic protein) was suspected to play a role in the etiology of autoimmune diseases and the gene expression ratio of IFN-γ/IL-4, and increased Golli-MBP was related to a lower ratio of IFN-γ/IL-4 in OLP patients.

A recent meta-analysis reported no significant differences in serum and salivary levels of IFN-γ between both OLP and control groups as well as erosive and non-erosive types in the saliva [28]. Zhou et al. [29] described the pathway of programmed death-1 (PD-1) and its ligand B7-H1, which may be implicated in OLP and have an important function in the negative modulation of T cell-mediated immune response, thus reducing serum IL-4 level in OLP. The Th1/Th2 cell imbalance may impact the OLP pathogenesis and elevate IL-4 level in OLP patients [23,30]. Malekzadeh et al. [23] suggested that Th1 cells were a dominant factor in cytokine secretion, but they could not be a causative agent and responsible for Th1/ Th2 cell imbalance. Two studies [12,31] showed a correlation between polymorphism and cytokine secretion.

Therefore, genetic factors (polymorphisms), the subtype of OLP, and other cytokines can affect the IFN-γ/IL-4 ratio. Yet, future studies are needed to focus on these effects and their association with one another. None of the studies analyzed in the meta-analysis reported the association of age and sex with the IFN-γ/IL-4 ratio. The researchers need to pay more attention to this issue in the future. Nevertheless, the present meta-analysis had several limitations, including low study sample size, different percentages of subtypes of OLP between the studies, high heterogeneity in the analyses, and lack of publication bias with strong results due to a few studies included in each analysis. In contrast to these limitations, the meta-analysis had two strengths, including high-quality studies and the stability of the results.

## 5. Conclusions

The results demonstrated that the serum and salivary ratio of IFN-γ/IL-4 cannot play a pivotal role in OLP development and severity. However, more studies in the future are needed to confirm this result with an emphasis on the factors involved in IFN-γ/IL-4 ratio.

## Figures and Tables

**Figure 1 medicina-55-00257-f001:**
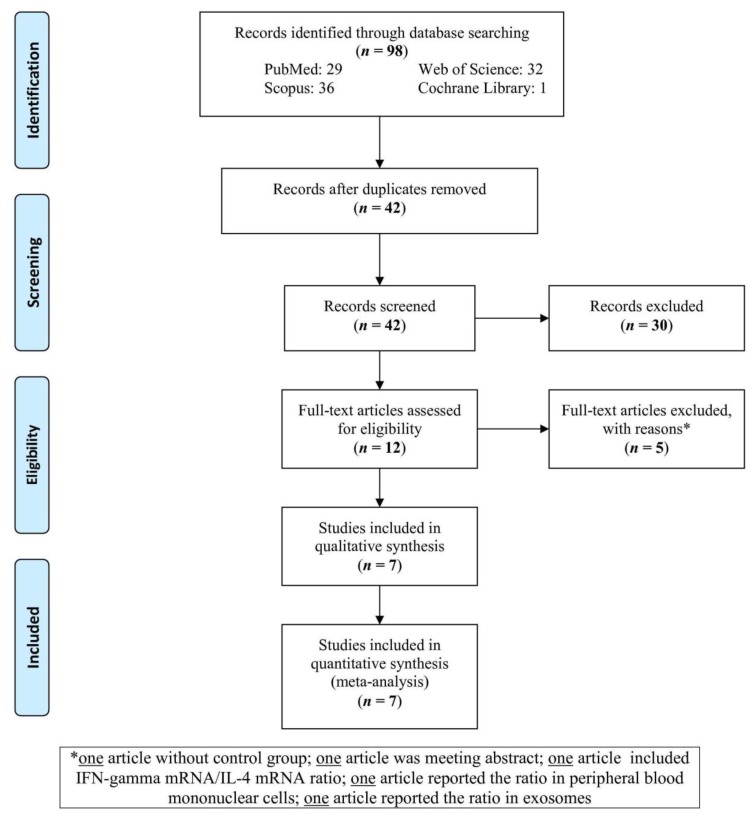
Flow-chart of the present study.

**Figure 2 medicina-55-00257-f002:**
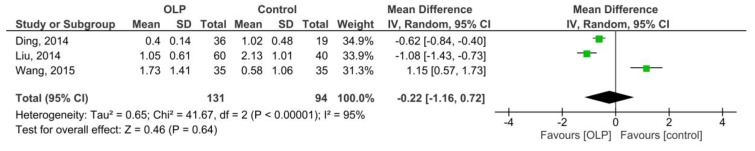
Forest plot of serum interferon-gamma/interleukin-4 (IFN-γ/IL-4) ratio in the patients with oral lichen planus compared to the controls.

**Figure 3 medicina-55-00257-f003:**
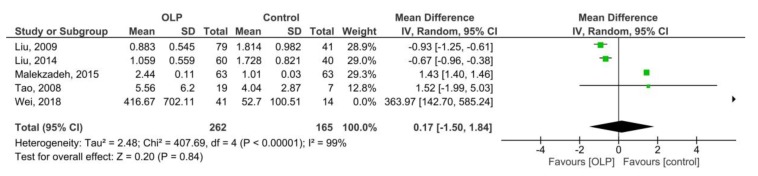
Forest plot of salivary IFN-γ/IL-4 ratio in the patients with oral lichen planus compared to the controls. Patients with erythematous/ulcerative subtype vs. reticular subtype (serum).

**Figure 4 medicina-55-00257-f004:**
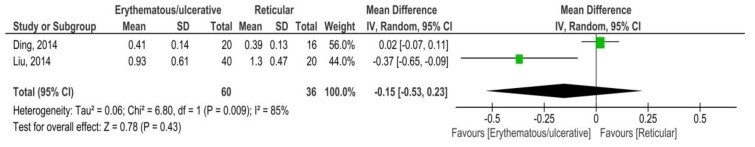
Forest plot of serum IFN-γ/IL-4 ratio in the patients with erythematous/ulcerative patients compared to reticular patients. Patients with erythematous/ulcerative subtype vs. reticular subtype (saliva).

**Figure 5 medicina-55-00257-f005:**
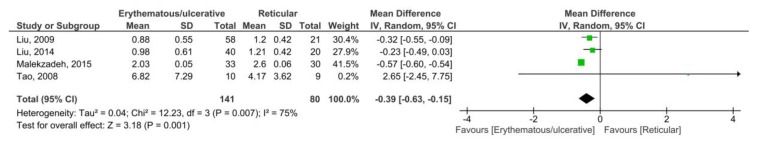
Forest plot of salivary IFN-γ/IL-4 ratio in the patients with erythematous/ulcerative patients compared to reticular patients.

**Table 1 medicina-55-00257-t001:** Characteristics of the studies included in the meta-analysis (*n* = 7).

The First Author, Year	Country	Mean Age (OLP/Control)	Male:Female (OLP/Control)	No. of OLP Patients	No. of Controls	Method	Sample
Tao, 2008 [19]	China	46.5/26.9	12:7/4:3	19	7	ELISA kit (eBioscience Inc., San Diego, CA, USA)	Saliva
Liu, 2009 [20]	China	46/41	37:42/20:21	79	41	ELISA kit (R&D Systems Inc., Minneapolis, MN, USA)	Saliva
Ding, 2014 [21]	China	45/43	15:21/7:12	36	19	ELISA kit (BioLegend, Inc., San Diego, CA, USA)	Serum
Liu, 2014 [22]	China	45/42	25:35/19:21	60	40	ELISA kit (R&D Systems Inc., Minneapolis, MN, USA)	Serum and Saliva
Malekzadeh, 2015 [23]	Iran	41.5/37	25:38/30:33	63	63	ELISA kit (eBioscience Inc., San Diego, CA, USA)	Saliva
Wang, 2015 [15]	China	53/54	4:31/4:31	35	35	ELISA kit (R&D Systems Inc., Minneapolis, MN, USA)	Serum
Wei, 2018 [4]	China	56.3/51.2	9:32/6:8	41	14	BD™ CBA Human Enhanced Sensitivity Flex Sets	Saliva

**Abbreviations**: OLP, oral lichen planus; ELISA, enzyme-linked immunosorbent assay; CBA, cytometric bead array.

**Table 2 medicina-55-00257-t002:** Sensitivity analysis for the pooled random-effects mean difference (MD) estimates in the subgroups.

Subgroup	Omitted Study	Removed Reason	Z	*p*	Heterogeneity	MD	95%CI (min, max)
OLP vs. Control (saliva)	Wei, 2018 [4]	Outlier study	0.17	0.86	99%	0.15	−1.51, 1.80
Erythematous/ulcerative vs. Reticular (saliva)	Tao, 2008 [19]	Outlier study	3.31	0.0009	81%	−0.40	−0.64, −0.16

**Table 3 medicina-55-00257-t003:** Quality assessment scores of the studies involved in the meta-analysis (*n* = 7) that every star means one point.

The First Author (year)	Selection	Comparability	Exposure	Total Points
Tao, 2008 [19]	***	*	***	7
Liu, 2009 [20]	****	**	***	9
Ding, 2014 [21]	***	**	***	8
Liu, 2014 [22]	***	*	***	7
Malekzadeh, 2015 [23]	***	**	***	8
Wang, 2015 [15]	***	**	***	8
Wei, 2018 [4]	***	*	***	7

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
