# Peer review of "Salivary and Serum Interferon-Gamma/Interleukin-4 Ratio in Oral Lichen Planus Patients: A Systematic Review and Meta-Analysis"

_medicina, 2019, doi:10.3390/medicina55060257_

Round 1
Reviewer 1 Report
The paper is well written and the topic is very interesting for oral medicine field.
Methodology is good for every step of the review.
I think that minor changes are needed to improve the quality of the paper and to make it easier to understand for the reader.
The funnel plot is not so easy to read and I think that forest plot are enough.
It is not clear in the discussion why you speak about some values as "elevated" and some others "decreased": decreased compared to what?
I think that it is necessary in the discussion to expand and highlight the importance of these findings for the clinical practice.
Author Response
1. The funnel plot is not so easy to read and I think that forest plot are enough.
Answer: Based on your valuable comment and also the comment of Reviewer 2, we deleted the funnel plots and added “and lack of a publication bias with strong results due to a few studies included in each analysis” in limitations.
2. It is not clear in the discussion why you speak about some values as "elevated" and some others "decreased": decreased compared to what?
Answer: “in OLP patients compared to controls” and “erythematous/ulcerative subtype compared to reticular subtype.”
3. I think that it is necessary in the discussion to expand and highlight the importance of these findings for the clinical practice.
Answer: Due to a few published studies about IFN-γ/IL-4 ratio in OLP patients, yet it has been not describe the importance of these finding in the clinical practice. But if the relationship between this ratio and the OLP is confirmed, this can be used as a future laboratory factor for the incidence or severity of this disease.
Reviewer 2 Report
From a methodological point of view, the manuscript is sound and could be accepted for publication. I have only a comment: to perform a meta-analysis with less than five studies is questionable, because all the parameters calculated in meta-analysis cannot be indicative. The Egger’s test seems particularly weak in this context.
Other points:
From the manuscript I understand that all the selected studies are controlled clinical trials. Being a Cross-sectional study is clearly an exclusion criterion. However, it is not clearly indicated in the manuscript. What does “case-control design” mean (page 2, line 82)? A case control study is generally a specific retrospective observational study.
Several evaluations were performed by a single author and it is undoubtedly a limitation.
Author Response
1. To perform a meta-analysis with less than five studies is questionable, because all the parameters calculated in meta-analysis cannot be indicative. The Egger’s test seems particularly weak in this context.
Answer: Based on your valuable comment and also the comment of Reviewer 1, we deleted the funnel plots and added “and lack of a publication bias with strong results due to a few studies included in each analysis” in limitations.
2. From the manuscript I understand that all the selected studies are controlled clinical trials. Being a Cross-sectional study is clearly an exclusion criterion. However, it is not clearly indicated in the manuscript. What does “case-control design” mean (page 2, line 82)? A case control study is generally a specific retrospective observational study.
Answer: We changed the sentences in Lines 84 & 85.
3. Several evaluations were performed by a single author and it is undoubtedly a limitation.
Answer: Thanks for your valuable comment and I will pay attention to this comment in the future studies. We had requested from other authors for “Quality Assessment” and “Data Extraction”. Because of similar results, I deleted second author and I thought that it is not necessary. Because of your valuable comment and its importance reducing bias to understand readers, we added the name of second author.
Reviewer 3 Report
Accepted in the present form. A well organized review paper.
Author Response
Thanks for your comments.
Reviewer 4 Report
I read with interest the manuscript entitled " Salivary and Serum Interferon-Gamma/Interleukin-4 Ratio in Oral Lichen Planus Patients: A Systematic Review and Meta-Analysis" authored by Mozaffari et al. The authors systematically reviewed the published reports and performed meta-analysis to assess the role of serum and salivary interferon-gamma/interleukin-4 ratio in the severity and development of Oral lichen planus (OLP). OLP is one of the most common inflammatory oral lesions that exhibit a potential for malignant transformation. Therefore, investigating potential therapeutic/mechanistic targets in this chronic lesion is considered advantageous. However, there are several concerns regarding the manuscript that should be addressed:
1) The text requires language revision. I recommend authors to seek proofreading from English-speaking colleague/service.
2) A total of 7 studies is rather low when it comes to a widely-studied lesion such as OLP. This has yielded a low study sample and high heterogeneity. The search strategy was limited to Scopus, Cochrane Library, PubMed/Medline, and Web of Science, however, it did not explore other international databases with broad subject areas, such as Embase SciELO, and Google Scholar etc. Please justify. Selection of a restricted subset of databases for conducting the literature search may compromise the results, or even lead to incorrect conclusions.
3) Did authors include only “full-text” studies in their search? This is not clear. If yes, this may result in a potential bias.
4) The introduction is short and does not appropriately cover the subject nor the clinical manifestations of OLP. In addition, OLP has been linked to several systemic diseases, so why did authors mention diabetes mellitus per se and excluded the rest.
5) In the search strategy, the starting date is missing.
6) Authors reported that quality evaluation was carried out by one author. For proper evaluation of reports, at least two reviewers are required per each article to reduce the number of errors, and yet, reviewers still often report different levels of bias for the same studies (Lensen et al., 2014).
7) In addition, in order to minimize errors, data extraction process is usually done by two reviewers who independently gather information from primary studies, and resolving disagreements with a third reviewer or by consensus. Please justify why only one reviewer was tasked for such vital processes.
8) Authors did not provide the relevant methodology on how they evaluated the publication bias in this manuscript. The importance of evaluating publication bias and the relevant methodologies for the assessment of meta-analysis bias can be found in (Soeken et al., 2003).
9) It is interesting that results showed no significant difference between groups except for the salivary ratio of IFN-γ/IL-4 in erythematous/ulcerative subtype compared to reticular subtype, the ratio being higher in reticular subtype than another subtype. Reticular OLP is the most common variant of the lesion and yet it is characterized by low/moderate immune response compared with other forms (and thus it may not require treatment). How could authors explain such finding in the clinical setting? Please highlight this in the discussion.
Author Response
1) The text requires language revision. I recommend authors to seek proofreading from English-speaking colleague/service.
Answer: Thanks for your valuable comments. We corrected some grammatical errors.
2) A total of 7 studies are rather low when it comes to a widely-studied lesion such as OLP. This has yielded a low study sample and high heterogeneity. The search strategy was limited to Scopus, Cochrane Library, PubMed/Medline, and Web of Science, however, it did not explore other international databases with broad subject areas, such as Embase SciELO, and Google Scholar etc. Please justify. Selection of a restricted subset of databases for conducting the literature search may compromise the results, or even lead to incorrect conclusions.
Answer: Thanks for your comment and I consider to this valuable comment in my future studies. We checked references of articles related to the subject for finding missed studies. There was no missed article and therefore we concluded to search these databases.
3) Did authors include only “full-text” studies in their search? This is not clear. If yes, this may result in a potential bias.
Answer: No. The search was done without any restriction.
4) The introduction is short and does not appropriately cover the subject nor the clinical manifestations of OLP. In addition, OLP has been linked to several systemic diseases, so why did authors mention diabetes mellitus per se and excluded the rest.
Answer: We added several sentences to introduction and corrected the sentence about “systemic disease”.
5) In the search strategy, the starting date is missing.
Answer: “from the database inception to March 2019”
6) Authors reported that quality evaluation was carried out by one author. For proper evaluation of reports, at least two reviewers are required per each article to reduce the number of errors, and yet, reviewers still often report different levels of bias for the same studies (Lensen et al., 2014).
Answer: Thanks for your valuable comment. We had requested from other authors for “Quality Assessment” and “Data Extraction”. Because of similar results, I deleted second author and I thought that it is not necessary. Because of your valuable comment and its importance reducing bias to understand readers, we added the name of second author.
7) In addition, in order to minimize errors, data extraction process is usually done by two reviewers who independently gather information from primary studies, and resolving disagreements with a third reviewer or by consensus. Please justify why only one reviewer was tasked for such vital processes.
Answer: Thanks for your valuable comment. We had requested from other authors for “Quality Assessment” and “Data Extraction”. Because of similar results, we thought not to write second author and it is not necessary. Because of your valuable comment and its importance reducing bias to understand readers, we added the name of second author.
8) Authors did not provide the relevant methodology on how they evaluated the publication bias in this manuscript. The importance of evaluating publication bias and the relevant methodologies for the assessment of meta-analysis bias can be found in (Soeken et al., 2003).
Answer: Based on the comments of Reviewers 1 and 2, we deleted the funnel plots and added “and lack of a publication bias with strong results due to a few studies included in each analysis” in limitations.
9) It is interesting that results showed no significant difference between groups except for the salivary ratio of IFN-γ/IL-4 in erythematous/ulcerative subtype compared to reticular subtype, the ratio being higher in reticular subtype than another subtype. Reticular OLP is the most common variant of the lesion and yet it is characterized by low/moderate immune response compared with other forms (and thus it may not require treatment). How could authors explain such finding in the clinical setting? Please highlight this in the discussion.
Answer: However, the meta-analysis showed no significant difference between groups except for the salivary ratio of IFN-γ/IL-4 in erythematous/ulcerative subtype compared to reticular subtype, the ratio being higher in reticular subtype than another. In addition, the analyses had a high heterogeneity that low studies included in each analysis can be one of the reasons of this heterogeneity. Therefore, the readers should pay attention to this point that the further studies are needed to prove this difference.
Round 2
Reviewer 4 Report
The authors have addressed my comments. Thank you!